# The Impact of Socio-Demographic Factors on the Functioning of Liver Transplant Patients

**DOI:** 10.3390/ijerph19074230

**Published:** 2022-04-01

**Authors:** Agnieszka Kisielska, Daria Schneider-Matyka, Kamila Rachubińska, Przemysław Ustanowski, Anita Rybicka, Elżbieta Grochans

**Affiliations:** 1Department of Infectious Diseases, Hepatology and Liver Transplantation, Pomeranian Medical University, 71-455 Szczecin, Poland; agnieszkakisielska@op.pl; 2Department of Nursing, Pomeranian Medical University in Szczecin, Żołnierska 48, 71-210 Szczecin, Poland; kamila.rachubinska@pum.edu.pl (K.R.); przemyslaw.ustianowski@pum.edu.pl (P.U.); anita.rybicka@pum.edu.pl (A.R.); grochans@pum.edu.pl (E.G.)

**Keywords:** non-adherence, liver transplantation, socio-demographic factors

## Abstract

(1) Background: The aim of this study was to evaluate the influence of socio-demographic factors and the time elapsed since liver transplantation on the functioning of patients after liver transplantation; (2) Methods: This is a survey-based prospective cohort study of 112 patients, performed using: The Inventory of Socially Supportive Behaviors (ISSB), The Acceptance of Illness Scale (AIS), the Beck Depression Inventory (BDI), and a questionnaire concerning sociodemographic data prepared using the Delphi method; (3) Results: Subjects under 40 years of age reported the highest social support. The longer the time since surgery, the lower the levels of adherence and support; (4) Conclusions: 1. In the study group, most support was received by women, people under 40 years of age, and those with secondary education. However, the level of social support decreased over time after the liver transplant operation. Patients who had undergone previous transplantation showed lower levels of adherence to therapeutic recommendations. 2. Patients who were in a relationship showed higher levels of illness acceptance than single ones. Women were more likely to experience depressive symptoms than men. 3. The time since liver transplantation is an important factor that affects patients’ functioning. This is a time when patients need more care, social support, and assistance in maintaining adherence to therapeutic recommendations.

## 1. Introduction

The introduction of immunosuppressive drugs to treat graft rejection has significantly advanced transplantation [1,2,3]. The optimal immunosuppressive treatment is considered to be one that achieves the stable function of the transplanted organ with minimal suppression of the immune system [4,5]. However, administration of immunosuppressive drugs often entails adverse effects, such as the increased risk of infections and neoplastic diseases, neuro- and nephrotoxicity, hypertension, hyperglycemia, lipidemia, anemia, leukopenia, thrombocytopenia, gingival hyperplasia, hair loss, intestinal disorders, mood disorders, muscular-articular pain, headache, as well as weight loss or gain [5,6,7,8]. The multitude of side effects of immunosuppressive therapy faced by patients after liver transplantation, as well as the patient’s views, lack of understanding of the purpose of treatment, poor knowledge of the disease and its treatment, belief about the harmfulness of drugs, limitation of cognitive functions, some personality traits, such as pessimism, forgetfulness, or disorganization may contribute to non-adherence. Demographic characteristics such as age, sex, and marital status are also important in the context of the functioning of transplant patients [9,10]. Evidence from the literature suggests that difficulties with treatment adherence among elderly people occur when self-management of the disease is required [11]. Social causes of non-adherence include a lack of support from family and friends and a lack of acceptance of the disease by the patient themselves or their environment [9,10]. Most studies confirm the impact of social support on patients’ quality of life, self-esteem, episodes of depression, and adaptation to living with the disease [12]. Adherence to therapeutic recommendations may be associated with adaptation *to disease-related* changes, and traits such as perseverance, optimism, belief in success, determination, and hope increase the chances of accepting the disease. Better adaptation to the disease means greater acceptance of the disease and less discomfort that it causes [13].

Most researchers agree that social support has a salutary effect on the mental health and well-being of an individual. Social support clearly plays a positive role in both maintaining health and supporting the healing process. Maintaining positive close relationships with others also helps to achieve balance in daily life, builds the individual’s sense of psychological well-being, plays an important role in adapting to critical events in life, attenuates the negative impact of stress on the sense of mental health, thus facilitating coping with stress [14,15,16,17,18,19,20].

Another important aspect of adherence to therapeutic recommendations by liver transplant patients is the acceptance of the disease, which means adopting a positive attitude towards a given situation or view favors the mobilization of the patient’s forces, and makes it possible to prevent a decrease in the quality of life. Acceptance of the disease creates a sense of security, reduces the intensity of negative emotions, and gives a sense of psychological comfort after organ transplantation, but also in the case of chronic diseases [21,22].

Gorevski et al. observed an increased prevalence of depression up to 23% before and 29% after organ transplantation [23]. It has been demonstrated that individuals showing depressive symptoms are less adherent to treatment recommendations [24]. This is a particularly important problem in patients after transplantation where strict adherence to recommendations, including those concerning multi-drug therapy, directly decides the patient’s life. According to the results of a meta-analysis on depression and mortality, the presence of depression increases the risk of death by 65% among transplant patients [25].

Many studies on adherence issues published in the medical literature report difficulties with cooperation between physicians, other medical professionals, and patients suffering from chronic conditions and requiring long-term therapy [26,27,28,29,30]. The underlying causes of nonadherence in long-term therapies are complex and can be divided according to the WHO report into those related to the patient, the disease, the therapy, the healthcare system, as well as social and economic factors [9,31,32]. In the present study, attention was paid to socio-demographic factors such as sex, age, place of residence, education, and marital status that may determine the functioning of patients after liver transplantation; the time elapsed since liver transplantation was also analyzed.

The purpose of this study was to evaluate the effect of sociodemographic factors and the time since liver transplantation on the functioning of patients after liver transplantation, taking into account social support, adherence to treatment, acceptance of the disease, and depression.

## 2. Materials and Methods

This is a survey-based prospective cohort study, which involved a group of 112 patients after liver transplantation who had a follow-up visit at the Transplant Outpatient Clinic of the Independent Public Regional Hospital in Szczecin or were hospitalized in the ward of infectious diseases, hepatology, and liver transplantation in the period when the study was conducted. The respondents were informed about the purpose of the research and gave their consent to take part in it. The study was approved by the Bioethics Committee of the Pomeranian Medical University in Szczecin, and was conducted in accordance with the principles of the Declaration of Helsinki. The study is a part of a larger research project.

Our investigation was carried out between August 2019 and February 2020. On the day of starting the research, the Transplant Outpatient Clinic provided medical care for 669 patients after liver transplantation, 36 of whom refused to participate in the study. The remaining patients who were not qualified for the study did not meet the study inclusion criteria. Patients with mental diseases, those with alcoholic cirrhosis, as well as hospitalized patients who could not participate in the study due to their health status were excluded.

The Delphi method was used to develop a tool to assess the level of adherence to therapeutic recommendations. The sources of data for this study were the recommendations [33,34,35] and the work of an expert panel. Six people were invited to the expert panel: a hepatologist-transplantologist, a hepatologist, a transplant surgeon, two nurses working in the transplant department, and one working in the hepatology outpatient clinic of the Independent Public Regional Hospital in Szczecin. The main criterion for recruitment to the team was the possession by the expert of at least five years’ work experience in transplantology in a hospital of a high referential level. The purpose of the expert panel was to develop a research instrument―the Adherence to Treatment Scale for Liver Transplant Patients. During the discussion, the experts presented their own observations on adherence to therapeutic recommendations. Based on the recommendations and brainstorming discussion, a pilot version of the scale was prepared and used in a preliminary qualitative study (semi-structured interview) of 10 patients who had undergone liver transplantation. The participants of the pilot study commented on the scope and quality of the wording of the scale. After the pilot study, the revised and supplemented questionnaire was discussed by the same team of experts. Finally, after a thorough analysis, 14 scale items were identified for which standards were established to determine the level of compliance with the recommendations. The items to be scored on the adherence scale were: a reason for transplantation, the time elapsed since surgery, drinking alcohol in any form, co-morbidities, taking medications other than those related to transplantation, number of pills taken per day, the frequency of medications taken per day, whether not taking the medications has happened, whether the physician has ever informed that test results may indicate irregular medication intake, whether all recommendations are followed, whether other sources of knowledge have been sought, self-assessed knowledge of the medications currently being taken, perceived adverse effects of treatment. Each item of the scale is scored on a two-point scale from 0 to 1, where 0 stands for adherence to therapeutic recommendations, and 1 means non-adherence. The following score ranges reflecting adherence to therapeutic recommendations were established: 0–4 points―high, 5–7 points―medium, 8–14 points―low. The author’s scale of adherence is a reverse scale, which means that the numerical scoring runs in the opposite direction―the higher the score, the lower the level of adherence.

Other research instruments used in the study were:The Inventory of Socially Supportive Behaviors (ISSB), used to analyze the types of social support (informational, emotional, instrumental, and evaluative). It contains 40 statements to be answered on a 5-point scale from 0—not at all, to 4—almost every day [36].The Acceptance of Illness Scale (AIS), used to determine the degree of acceptance of the disease by the patient. The questionnaire consists of eight statements concerning the negative consequences of poor health. The answers are weighted on a scale from 1 to 5. The higher the score, the higher the adaptation to disease-related limitations [37].The Beck Depression Inventory-II (BDI-II) is a self-descriptive tool used to assess the severity of depressive symptoms. It consists of 21 questions with 4 response options, which are scored from 0 to 3. The score was obtained by summing the point values corresponding to each statement. The results were interpreted by referring to the standardized ranges, where: 0–13 means no depression or minimal depressive symptoms, 14–19―mild depression, 20–28―moderate depression, and 29–63―severe depression [38].A questionnaire concerning socio-demographic data, i.e., age, sex, place of residence, education, and marital status.

### The Methods of Descriptive Statistics Were Used for Statistical Analysis

Depending on the type of variable, the following were used: mean and standard deviation and the structure indicators (frequency and percentage). The dependent variable expressed on the metric scale was the result obtained from the measurement of standardized tools with confirmed psychometric properties. In the case of comparing a larger number of samples (groups), a one-way analysis of variance (ANOVA) with the post hoc least significant difference (LSD) test was used. The effect size was calculated using the coefficient η^2^. For the analysis of differences between two samples, evaluation was conducted using the parametric Student’s *t*-test. The r-Pearson correlation coefficient was determined to estimate the relationship between the two metric variables. Statistical software STATISTICA version 13.3 (TIBCO Software Inc., Szczecin, Poland) was used for statistical calculations. Statistical hypotheses were verified at a predetermined significance level of 0.05.

## 3. Results

Women constituted 50.9% of the study sample. Of the sample, 48.2% were patients between 40 and 60 years of age, 32.1% were above 60 years of age, and 19.6% were under 40 years of age. In the group of patients below 40 years of age, the youngest patient was 20 years old, and in the group of patients above 60 years of age, the oldest patient was 74 years old. The mean age was 51.33 years. Most respondents (74.1%) lived in urban and 25.9% in rural areas. Secondary education was declared by 42.9%, primary or vocational by 37.6%, and higher education by 19.6%. Of the respondents, 36.6% were single and 63.4% were in a relationship. More than half of the respondents (55.4%) were patients more than two years after liver transplantation, more than 35.7% of the respondents were less than one year after the transplantation. The smallest group consisted of patients whose transplantation was performed between 1 and 2 years before the survey (Table 1).

Most respondents (54.5%) showed an average level of adherence to therapeutic recommendations, 34.8% were characterized by low adherence, and only 10.7% of the respondents thoroughly followed the recommendations.

Analysis of the severity of depression according to the BDI revealed that the majority of the respondents had no or low depressive symptoms (55.2%), 19% had moderate depressive symptoms, 13.8% were mildly depressed, and 12.1% were severely depressed.

On average, adherence to therapeutic recommendations according to our Adherence to Treatment Scale for Liver Transplant Patients was at a medium level (6.8 ± 1.85). The average value of received emotional support was 29.2 ± 10.81, informational support—30.1 ± 13.76, instrumental support—32.1 ± 13.88, and evaluative support—4.2 ± 6.04. The respondents declared a rather high level of disease acceptance according to the AIS (M = 27.5 ± 8.03). The average BDI score was 9.3 ± 8.97, which stands for no or mild depressive symptoms. The incidence of anxiety was on an average level—both anxiety as a state (M = 5.3 ± 2.13) and anxiety as a trait (M = 5.2 ± 2.08).

The data analysis revealed that sex was a factor differentiating transplant patients in terms of emotional support (t(109) = 4.011; *p* < 0.001; dCohen = 0.76; 95%CI [0.38–1.15]). A mean level of such support was higher for women than for men (32.86 vs. 25.13). In the case of informational support the situation looked similar (t(109) = 3.013; *p* = 0.003; dCohen = 0.57; 95%CI [0.19–0.95]) (33.67 for women vs. 26.06 for men). Additionally, in the case of instrumental support (t(109) = 3.229; *p* = 0.002; dCohen = 0.61; 95%CI [0.23–0.99]) and evaluative support (t(109) = 3.126; *p* = 0.002; dCohen = 0.59; 95%CI [0.21–0.97]), sex was a differentiating factor among transplant patients. Higher mean levels of support were reported by females compared to males for both instrumental support (36.19 vs. 28.00) and evaluative support (15.82 vs. 12.37).

Sex differentiated transplant patients also in terms of depression (t(109) = 2.048; *p* = 0.043; dCohen = 0.39; 95%CI [0.01–0.77]). A higher mean level of depression was noted for women compared to men (11.05 vs. 7.61). There were no significant differences in the level of adherence to treatment recommendations and acceptance of the disease between women and men (Table 2).

Analysis of the influence of age on social support according to the ISSB showed statistically significant differences (*p* < 0.05) in all categories of support. The highest mean level of emotional support (F(2.109) = 3.352; *p* = 0.039; η^2^ = 0.058) was noted for the youngest people (under 40 years of age), and it was significantly higher compared to middle-aged people (40–60 years of age; the LSD test, *p* = 0.019) and elderly people (over 60 years of age; the LSD test, *p* = 0.021). In contrast, there was no difference in the mean level of emotional support between the groups of 40–60-year-olds and over 60-year-olds (the LSD test, *p* = 0.897). Age was a differentiating factor in terms of informational support (F(2, 109) = 4.402; *p* = 0.014; η^2^ = 0.075). The youngest individuals (under 40 years of age) reported the highest mean level of informational support, and it was significantly higher compared to middle-aged individuals (40–60 years; the LSD test, *p* = 0.006) and elderly patients (over 60 years; the LSD test, *p* = 0.010). In contrast, there were no differences in mean levels of informational support between the groups of 40–60-year-olds and over 60-year-olds (the LSD test, *p* = 0.967). Age differentiated transplant patients also in terms of instrumental support (F(2, 109) = 5.195; *p* = 0.007; η^2^ = 0.087)―the highest mean level of instrumental support was noted for the youngest individuals (under 40 years of age), and it was significantly higher compared to middle-aged individuals (40–60 years; the LSD test, *p* = 0.002) and elderly subjects (over 60 years; the LSD test, *p* = 0.008). There were no differences in mean instrumental support between the groups of 40–60-year-olds and over 60-year-olds (the LSD test, *p* = 0.820). Significant differences were found in the level of evaluative support (F(2, 109) = 3.287; *p* = 0.041; η^2^ = 0.057) depending on age. The youngest people (under 40 years of age) reported the highest mean level of such support, and it was significantly higher compared to middle-aged people (40–60 years of age; the LSD test, *p* = 0.015) and elderly people (over 60 years of age; the LSD test, *p* = 0.033). At the same time, there was no difference in the mean level of evaluative support between the groups of 40–60-year-olds and over 60-year-olds (the LSD test, *p* = 0.845).

There were no statistically significant differences (*p* > 0.05) in the level of adherence to treatment recommendations, the degree of acceptance of the disease according to the AIS, and the severity of depression according to the BDI between the age groups (Table 3).

The study showed no significant effect of place of residence on the level of adherence to therapy, the received support, or psychological and illness-related variables (Table 4).

Education differentiated post-transplant patients in terms of the levels of emotional, informational, and evaluative support received.

The highest mean level of emotional support (F(2, 109) = 6.129; *p* = 0.003; η^2^ = 0.101), was reported by those with secondary education, and it was significantly higher compared to those with primary/vocational education (the LSD test, *p* = 0.002) and those with higher education (the LSD test, *p* = 0.014). There was no difference in the mean level of emotional support between the primary/vocational education and higher education groups (the LSD test, *p* = 0.895).

Education was a factor differentiating transplant patients in terms of informational support received (F(2, 109) = 3.161; *p* = 0.046; η^2^ = 0.055). The highest mean level of informational support was recorded for those with secondary education; it was significantly higher compared to those with primary/vocational education (the LSD test, *p* = 0.018). There were no differences in the mean levels of informational support between people with primary/vocational and higher education (the LSD test, *p* = 0.725), and between people with secondary and higher education (the LSD test, *p* = 0.110).

Differences were also found in evaluative support (F(2, 109) = 4.071; *p* = 0.020; η^2^ = 0.070). The highest mean level of such support was noted for people with secondary education, and it is significantly higher compared to people with primary/vocational education (the LSD test, *p* = 0.009). There were no differences in the mean levels of evaluative support between people with primary/vocational and higher education (the LSD test, *p* = 0.769), and between people with secondary and higher education (the LSD test, *p* = 0.058). Education had no significant impact on adherence, instrumental support, or other psychological and health-related variables (Table 5).

When analyzing the effect of marital status on individual variables, it was found to be a factor differentiating transplant patients in terms of illness acceptance (t(109) = 2.344; *p* = 0.021; dCohen = 0.46; 95%CI [0.07–0.85]). Patients who were in a relationship had a higher mean level of illness acceptance than single ones (28.79 vs. 25.17). Marital status had no impact on other variables (Table 6).

Additionally, the influence of time elapsed since liver transplantation on adherence to therapy, social support according to the ISSB, acceptance of disease according to the AIS, and depression according to the BDI was analyzed. Statistically significant differences (*p* < 0.05) were found in the levels of informational and evaluative support. The highest mean level of informational support (F(2, 109) = 3.302; *p* = 0.041; η^2^ = 0.057) was noted for those being less than one year after transplantation, and it was significantly higher compared to those who were more than two years after surgery (the LSD test, *p* = 0.012). No difference in the mean level of informational support was found between those who were one to two years after transplantation and those who were more than two years after transplantation (the LSD test, *p* = 0.626). There were no differences in the mean levels of informational support between the groups of those who were less than one year after transplantation and those who were one to two years after transplantation (the LSD test, *p* = 03.19). The time since transplantation differentiated patients in terms of evaluative support (F(2, 109) = 4.662; *p* = 0.011; η^2^ = 0.079). Those who were less than one year after transplantation had the highest mean level of evaluative support, and it was significantly higher compared to those who were more than two years after transplantation (the LSD test, *p* = 0.003). In contrast, there was no difference in the mean level of evaluative support between those who were one to two years after transplant surgery compared to those being more than two years after transplantation (the LSD test, *p* = 0.747). There was also no difference in the mean level of evaluative support between those who were less than one year and those who were one to two years after transplantation (the LSD test, *p* = 0.157). A statistically significant relationship was found between the time elapsed since liver transplantation and adherence to therapy. The longer the time since surgery, the lower the level of adherence (r = 0.31; t = 3.346; *p* = 0.001). There was a positive correlation (r = 0.31), although the interpretation shows a negative direction because the adherence scale is a reverse scale. No statistically significant differences were observed in the case of the other variables (Table 7).

A statistically significant relationship was found between the time elapsed since liver transplantation and the level of adherence to therapeutic recommendations. The longer the time since surgery, the lower the level of adherence (r = 0.31; t = 3.346; *p* = 0.001). The numerical correlation was positive (r = 0.31), although the interpretation shows a negative direction because the adherence scale is a reverse scale. There was also a statistically significant negative correlation between the time elapsed since surgery and the levels of: emotional support (r = −0.26; t = −2.779; *p* = 0.006), informational support (r = −0.31; t = −3.275; *p* = 0.001), instrumental support (r = −0.28; t = −2.916; *p* = 0.004), and evaluative support (r = −0.35; t = −3.837; *p* < 0.001). Support in each of the above mentioned categories decreased with the number of months since surgery (Table 8).

## 4. Discussion

The procedure of qualification for organ transplantation involves the recognition of risk factors for *nonadherence*. The risk factors for nonadherence are believed to be: young age, low education level, a long distance from the transplant center, and lack of emotional support [39].

On the other hand, the first study comparing older and younger liver recipients in medical records demonstrated that adult liver recipients over 65 years of age showed higher adherence compared to younger ones [40]. A similar analysis performed by Casleberry et al. among lung recipients revealed that non-adherence was significantly more common among patients between 18 and 20 years of age and those with a lower level of education than among patients between 21 and 50 years of age and those with higher education [41]. In a cross-sectional study of 101 renal transplant patients in Taiwan, age was the only predictor of health care adherence [42].

In our study, a relationship between the age of liver recipients and the support received was observed. The youngest subjects (under 40 years of age) recorded the highest mean levels of emotional, informational, instrumental, and evaluative support received. At the same time, age had no effect on the level of adherence to therapeutic recommendations. This may be due to a significantly high level of support provided for the youngest patients. It is worth noting that due to a small group of respondents under 20 years of age in our study, we accepted ages up to 40 years as the upper age limit for young recipients, which also may have affected the results.

The results of studies on the influence of sex on adherence to therapeutic recommendations are not unequivocal. Some research authors have demonstrated lower adherence to treatment recommendations after organ transplantation among men [43,44,45], patients with low income, and those with higher or secondary education [45]; however, other findings indicate female sex as a risk factor for non-adherence [46,47]. In an analysis by Marsicano et al., younger age and male sex were not identified as important contributors to non-adherence [44]. In a Polish study analyzing health behaviors of 115 adult liver recipients with varying times since transplantation, high discipline in adherence to therapeutic recommendations was observed. In this group, 93.9% of the respondents reported taking immunosuppressive drugs regularly at the same time each day, and 64.3% did not take any over-the-counter drugs without consultation. The study indicated that some sociodemographic variables, such as female sex, higher education, and receiving a pension were associated with significantly wider adoption of healthy behavior patterns. Rural residents chose healthy behaviors less often than their counterparts living in urban areas [48]. These observations are consistent with the results showing that patients who adhere to medical recommendations care more about leading a healthy lifestyle, diet, and physical activity. They also have regular medical check-ups and participate in preventive screenings [49].

In our study, sex had no effect on adherence, but it was a factor differentiating transplant patients in terms of emotional, informational, instrumental, and evaluative support. Women reported higher mean levels of all types of support.

In a study by Hugon et al., living alone was a predictor of post-transplant non-adherence in multivariate analysis [50]. Martynów et al. informed that patients with a high level of illness acceptance were more likely to adhere to treatment recommendations. Correlation analysis showed a statistically significant, strong association between the level of treatment adherence and acceptance of disease according to the AIS [51].

When analyzing the results of our study, we found that marital status was a factor differentiating transplant patients in terms of illness acceptance. People who were in a relationship showed a higher mean level of illness acceptance.

Stilley et al., who examined correlates and predictors of adherence to immunosuppressive drug regimens among 152 liver recipients, found no relationship between treatment adherence and sociodemographic factors, such as age, sex, marital status, education, or income level [52]. Another study concerning factors associated with adherence to treatment after organ transplantation provided evidence that poorer treatment adherence on self-report scales was related to young age, residence in a town without metro access, and having six or more comorbidities [53]. In their correlation analysis of socio-demographic data, social support, depression, anxiety, and pre-transplant adherence, Dobbels et al. found that a higher education level, lower social support, and lower conscientiousness were independent predictors of non-adherence to recommendations [54].

In our study, education differentiated post-transplant patients in terms of the level of emotional, informational, and evaluative support received. Education had no significant effect on the level of adherence, instrumental support, or other psychological and health-related variables.

In the search for determinants of treatment non-adherence, it has been found that adherence to therapeutic recommendations decreases over time [55]. Researchers analyzing transplant patients observed a relationship between the time elapsed since surgery and poorer adherence to therapy. Studies focused on treatment adherence among kidney recipients have demonstrated that adherence to an immunosuppressive regimen decreases with time from the surgery [56,57]. Similar results were obtained in a study by Rodrigue et al. among liver transplant patients. The rates of immunosuppressive drug non-use and drug holidays during the first two years after liver transplantation are unacceptably high. Additionally, pre-transplant mood disorders and instability of social support increase the risk of non-adherence [58]. Studies concerning correlations between the time elapsed since the procedure and the levels of various types of support provide inconsistent results [59].

Our study showed a statistically significant relationship between the time elapsed since liver transplantation and the level of treatment adherence. The longer the time since surgery, the lower the level of adherence. Additionally, it was shown that social support also decreased with the number of months from the surgery. A statistically significant negative correlation was found between the time elapsed since the surgery and the levels of emotional support, informational support, instrumental support, and evaluative support, which may translate into poorer adherence to therapeutic recommendations.

Both our results and those of other authors suggest that sociodemographic factors and the time since transplantation may influence adherence to therapy.

The presented research results confirm the statistical significance, which excludes the probability of randomness, however, it does not prove the clinical significance, which focuses on the possibility of applying the findings in practice or the direct impact of the findings on the patient. Nevertheless, research results may indicate which factors and spheres of functioning of liver transplant patients increase the risk of nonadherence. The identification of risk factors would allow the creation of a profile of patients who will be predisposed to nonadherence by their sociodemographic situation. The presented issues require further research with a larger study sample and a control group included as well as other transplant centers. This would allow for the formulation of generalized conclusions that go beyond the studied group of patients after liver transplantation. The lack of a control group and monocentricity are important limitations of this study.

## 5. Conclusions

In the study group, most support was received by women, people under 40 years of age, and those with secondary education. However, the level of social support decreased over time after the liver transplant operation. Patients who had undergone previous transplantation showed lower levels of adherence to therapeutic recommendations.Patients who were in a relationship showed higher levels of illness acceptance than single ones. Women were more likely to experience depressive symptoms than men.The time since liver transplantation is an important factor that affects patients’ functioning. This is a time when patients need more care, social support, and assistance in maintaining adherence to therapeutic recommendations.

## Figures and Tables

**Table 1 ijerph-19-04230-t001:** The structure of age, sex, and place of residence of the study group.

Variable	
Sex	female
	male
	gaps
Age group	<40 y.o.
	40–60 y.o.
	>60 y.o.
Place of residence	village
	city
Education	primary/vocational education
	secondary
	higher
Marital status	in a relationship
	single
Time since surgery	<1 year
	1 to 2 years
	>2 years

n—number of cases, %—percentage of the total study group.

**Table 2 ijerph-19-04230-t002:** Adherence to therapy, social support according to the ISSB, acceptance of disease according to the AIS, and depression according to the BDI among the liver transplant patients by sex.

Variable	Woman (*n =* 57)		Men (*n =* 54)		t	*p* *
	M	SD	M	SD		
Level of adherence to treatment recommendations	6.82	1.82	6.80	1.92	0.080	0.937
Social support						
*Emotional support*	32.86	9.69	25.13	10.62	4.011	**<0.001**
*Informational support*	33.67	13.67	26.06	12.91	3.013	**0.003**
*Instrumental support*	36.19	14.82	28.00	11.62	3.229	**0.002**
*Evaluative support*	15.82	5.88	12.37	5.75	3.126	**0.002**
Acceptance of illness	27.19	8.66	27.70	7.44	–0.332	0.740
Depressiveness	11.05	10.01	7.61	7.42	2.048	**0.043**

M—mean, SD—standard deviation, *—Student’s *t*-test.

**Table 3 ijerph-19-04230-t003:** Adherence to treatment recommendations, social support according to the ISSB, acceptance of disease according to the AIS, and depression according to the BDI among liver transplant patients by age.

Variable	<40 Years (*n =* 22)		40 to 60 Years (*n =* 54)		>60 Years (*n =* 36)		F	*p* *
	M	SD	M	SD	M	SD		
Level of adherence to treatment recommendations	6.45	1.68	6.85	1.92	6.97	1.87	0.552	0.577
Social support								
*Emotional support*	34.41	9.24	28.02	11.22	27.72	10.34	3.352	**0.039**
*Informational support*	37.64	14.89	28.26	12.58	28.14	13.53	4.402	**0.014**
*Instrumental support*	40.36	13.96	29.87	13.16	30.53	13.35	5.195	**0.007**
*Evaluative support*	17.09	6.32	13.39	5.64	13.64	6.10	3.287	**0.041**
Acceptance of illness	27.45	9.05	28.20	7.34	26.36	8.46	0.564	0.570
Depressiveness	10.23	10.91	9.72	8.22	8.11	8.90	0.489	0.615

M—mean, SD—standard deviation, *—single-factor ANOVA.

**Table 4 ijerph-19-04230-t004:** Adherence to treatment, social support according to the ISSB, acceptance of disease according to the AIS, and depression according to the BDI among liver transplant patients by place of residence.

Variable	Village (*n =* 29)		City (*n =* 83)		t	*p* *
	M	SD	M	SD		
Level of adherence to treatment recommendations	6.59	1.97	6.89	1.81	–0.763	0.447
Social support						
*Emotional support*	32.41	10.10	28.05	10.87	1.895	0.061
*Informational support*	33.45	13.97	28.88	13.57	1.549	0.124
*Instrumental support*	33.83	12.91	31.55	14.22	0.758	0.450
*Evaluative support*	14.76	6.87	14.00	5.76	0.580	0.563
Acceptance of illness	27.79	8.70	27.35	7.83	0.255	0.799
Depressiveness	9.31	8.98	9.30	9.02	0.005	0.996

M—mean, SD—standard deviation, *—Student’s *t*-test.

**Table 5 ijerph-19-04230-t005:** Adherence to treatment recommendations, social support according to the ISSB, acceptance of disease according to the AIS, and depression according to the BDI among liver transplant patients by education.

Variable	Primary/Vocational Education (*n =* 42)		Secondary (*n =* 48)		Higher (*n =* 22)		F	*p* *
	M	SD	M	SD	M	SD		
Level of adherence to treatment recommendations	6.67	1.96	6.94	1.67	6.82	2.08	0.236	0.790
Social support								
*Emotional support*	26.10	11.06	33.13	9.75	26.45	10.15	6.129	**0.003**
*Informational support*	26.88	12.97	33.73	13.39	28.14	14.70	3.161	**0.046**
*Instrumental support*	28.81	12.75	35.73	14.00	30.68	14.41	3.046	0.052
*Evaluative support*	12.69	5.78	16.02	5.55	13.09	6.75	4.071	**0.020**
Acceptance of illness	26.26	8.06	27.92	7.78	28.77	8.56	0.837	0.436
Depressiveness	10.88	9.35	8.63	8.44	7.77	9.30	1.109	0.334

M—mean, SD—standard deviation, *—single-factor ANOVA.

**Table 6 ijerph-19-04230-t006:** The level of adherence to treatment recommendations, social support according to the ISSB, acceptance of disease according to the AIS, and depression according to the BDI among liver transplant patients by marital status.

Variable	In a Relationship (*n =* 71)		Single (*n =* 41)		t		*p* *
	M	SD	M	SD			
Level of adherence to treatment recommendations	6.76	1.92	6.90	1.74	–0.389		0.698
Social support							
*Emotional support*	29.35	10.95	28.88	10.67	0.223		0.824
*Informational support*	30.34	13.98	29.59	13.53	0.278		0.782
*Instrumental support*	32.07	13.72	32.27	14.32	–0.072		0.942
*Evaluative support*	13.99	6.27	14.56	5.69	–0.483		0.630
Acceptance of illness	28.79	8.18	25.17	7.29	2.344	**0.021**	
Depressivenes	8.24	8.29	11.15	9.88	–1.665	0.099	

M—mean, SD—standard deviation, *—Student’s *t*-test.

**Table 7 ijerph-19-04230-t007:** The level of treatment adherence, social support according to the ISSB, disease acceptance according to the AIS, and depressiveness according to the BDI among liver transplant patients by the time since liver transplantation.

Variable	<1 Year (*n =* 41)		1 to 2 Years (*n =* 10)		>2 Years (*n =* 61)		F	*p* *
	M	SD	M	SD	M	SD		
Level of adherence to treatment recommendations	6.33	1.79	6.40	2.32	7.19	1.75	3.052	0.051
Social support								
*Emotional support*	31.35	10.59	31.00	11.48	27.48	10.71	1.735	0.181
*Informational support*	34.38	13.89	29.60	14.05	27.35	13.13	3.302	**0.041**
*Instrumental support*	35.05	15.26	33.20	14.29	30.10	12.70	1.598	0.207
*Evaluative support*	16.45	5.57	13.50	7.00	12.85	5.84	4.662	**0.011**
Acceptance of illness	26.48	7.22	26.30	7.41	28.29	8.62	0.734	0.483
Depressivenes	9.33	9.28	10.20	5.98	9.15	9.27	0.059	0.943

M—mean, SD—standard deviation, *—single-factor ANOVA.

**Table 8 ijerph-19-04230-t008:** The correlation between the time elapsed since liver transplantation and the levels of adherence to therapeutic recommendations, social support according to the ISSB, acceptance of disease according to the AIS, depression according to the BDI, and anxiety according to the STAI.

Variable—Time Elapsed Since Liver Transplantation	r-Pearsona	t	*p*
Variable—time elapsed since liver transplantation	0.31	3.346	**0.001**
Social support			
*Emotional support*	–0.26	–2.779	**0.006**
*Informational support*	–0.31	–3.275	**0.001**
*Instrumental support*	–0.28	–2.916	**0.004**
*Evaluative support*	–0.35	–3.837	**<0.001**
Acceptance of illness	0.07	0.664	0.508
Depressivenes	0.03	0.282	0.778

r-Pearson—Pearson’s linear correlation coefficient, t—value of test statistic, *p*—test probability.

## Data Availability

Data sharing not applicable.

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
