# Peer review of "The Impact of Socio-Demographic Factors on the Functioning of Liver Transplant Patients"

_ijerph, 2022, doi:10.3390/ijerph19074230_

Round 1

Reviewer 1 Report

Dear authors,

I have reviewed your paper and have the following comments:

Major comments

a. The abstract needs to be rewritten. It does not include the type of the study (prospective, retrospective, comparative or not etc) and contains unnecessary repetition (lines 13-14). Presentation and significance of results is unclear to the reader ("the highest mean level of received social support"?) and this is not alleviated by adding statistical parameters. Statistical significance does not mean clinical significance.

b. The whole paper needs substantial improvement in terms of style and English. Some examples below, but these are present throughout the paper really:

-line 28-29: transplantation is a surgical procedure that involves transplantation

-line 31: salvation used as rescue therapy (I assume)

-line 32: people can save themselves

-line 43: with which people with liver transplantation struggle

And so on.

c. Introduction can be shortened (not sure the readers need to be told what a transplant is). It is not clear whether the information provided is specific to liver transplant (does it apply to other transplants as well?). Line 87 should explain what WHO report is meant. Text on the present study do not belong to the introduction.

d. Material and methods: There is no clear description of the study type and of inclusion and exclusion criteria. There is no description of the endpoints. The criteria used for the selection of experts did not take into account their previous experience with working with scales/scores. The developed scale does not seem to be validated and therefore study results are questioned. This cannot be mitigated by the abundance of statistical figures, which seems inadequate for the design of the study. The scale parameters need to be reviewed (line 131 "whether I (?) follow"...)

e. Results: The formatting needs to be redone so that the text can be easier read. There are too many statistical parameters listed for a small non-comparative study and the true significance of the results (beyond statistical significance) is fully unclear.

f. Discussion: this section should focus on what is perceived important in this study. As it stands the discussion is very difficult to follow.

g. Conclusion: There is no general conclusion about the usefulness of these results for the medical community and for the patients and it is not clear what this study adds to the existing knowledge.

Author Response

March 15, 2022

Dear Sir or Madam,

We are very grateful for the review of our article titled “The impact of socio-demographic factors on the functioning of liver transplant patients”.

We would like to thank you for all your comments and suggestions, which helped us to improve our manuscript.

The following corrections have been introduced in order to address the suggestions of the Reviewer 1:

Dear authors,

I have reviewed your paper and have the following comments:

Major comments

  1. The abstract needs to be rewritten. It does not include the type of the study (prospective, retrospective, comparative or not etc.) and contains unnecessary repetition (lines 13-14). Presentation and significance of results is unclear to the reader ("the highest mean level of received social support"?) and this is not alleviated by adding statistical parameters. Statistical significance does not mean clinical significance.

The abstract has been modified according to the Reviewer’s instructions.

  1. The whole paper needs substantial improvement in terms of style and English. Some examples below, but these are present throughout the paper really:

-line 28-29: transplantation is a surgical procedure that involves transplantation

-line 31: salvation used as rescue therapy (I assume)

-line 32: people can save themselves

-line 43: with which people with liver transplantation struggle

And so on.

The paper has been carefully proofread and extensively revised in terms of the language.

  1. Introduction can be shortened (not sure the readers need to be told what a transplant is). It is not clear whether the information provided is specific to liver transplant (does it apply to other transplants as well?). Line 87 should explain what WHO report is meant. Text on the present study do not belong to the introduction.

The introduction has been shortened, and passages that the reviewer suggested should be removed have been deleted. Missing information regarding cited research results from other authors has been completed. Information regarding the WHO report on nonadherence in long term therapies has been supplemented.

  1. Material and methods: There is no clear description of the study type and of inclusion and exclusion criteria. There is no description of the endpoints. The criteria used for the selection of experts did not take into account their previous experience with working with scales/scores. The developed scale does not seem to be validated and therefore study results are questioned. This cannot be mitigated by the abundance of statistical figures, which seems inadequate for the design of the study. The scale parameters need to be reviewed (line 131 "whether I (?) follow"...)

Information regarding the study type has been completed - this is a cohort prospective study. The endpoint is to achieve maximum comfort in psychosocial functioning by achieving disease acceptance, maintaining social support, proper treatment adherence, and minimizing depressive symptoms in patients after liver transplantation.

Information regarding study inclusion and exclusion criteria was completed.

The Delphi method was used for the sake of the study with the participation of a panel of experts, whose qualifications are given in the description of methods section. The four authors of the manuscript were also panel participants, and it was they who constructed the questionnaire in collaboration with the experts, modifying it according to their comments and the results of the pilot study. These authors of the manuscript have vast experience in conducting research based on diagnostic survey with the use of research tools (survey questionnaires). According to the assumptions of the Delphi method the tool does not need to be validated, because it is the method that gives the research credibility.

The items to be scored on the adherence scale were: a reason for transplantation, the time elapsed since surgery, drinking alcohol in any form, co-morbidities, taking medications other than those related to transplantation, number of pills taken per day, the frequency of medications taken per day, whether not taking the medications has happened, whether the physician has ever informed that test results may indicate irregular medication intake, whether all recommendations are followed, whether other sources of knowledge have been sought, self-assessed knowledge of the medications currently being taken, perceived adverse effects of treatment.

  1. Results: The formatting needs to be redone so that the text can be easier read. There are too many statistical parameters listed for a small non-comparative study and the true significance of the results (beyond statistical significance) is fully unclear.

Thank you for pointing out the need to highlight the difference between statistical significance and clinical significance of the studies presented. The authors have addressed this issue in the summary as limitations of the research.

  1. Discussion: this section should focus on what is perceived important in this study. As it stands the discussion is very difficult to follow.

The discussion has been modified and reorganized to increase the clarity and readability of this section of the manuscript. Reduntant information has been removed.

  1. Conclusion: There is no general conclusion about the usefulness of these results for the medical community and for the patients and it is not clear what this study adds to the existing knowledge.

The conclusions were rewritten according to the Reviewer’s instructions.

Kindest regards

Daria Schneider-Matyka

Reviewer 2 Report

In this study the authors aim to evaluate in transplanted patients influence of socio-demographic factors and the time elapsed since liver transplantation on the functioning of patients.

The abstract and paper are difficult to read and need to be more concise and more flowing.

The research design is quite appropriate, and methods adequately described, but the monocentric design of study represent a limitation.

There are for me several points which need more clarification:

  • the authors should clarify the time frame for the study enrollment;
  • the authors should clarify the exclusion criteria of the study and the number of transplanted patients who refused to be tested (they could be the patients with less compliant) ;
  • The etiology of hepatic disease, especially alcoholic cirrhosis, before the transplantation might play a role in this setting and should be add in the analysis;
  • Are there patients with a psychiatric diagnosis? Or are these patients excluded from the study?
  • Authors did not clarify how the research can have an impact on the clinical management of this patients.
  • The authors should add in the conclusions the limitations of the study: monocentric, no control group etc.

Author Response

March 15, 2022

Dear Sir or Madam,

We are very grateful for the review of our article titled “The impact of socio-demographic factors on the functioning of liver transplant patients”.

We would like to thank you for all your comments and suggestions, which helped us to improve our manuscript.

The following corrections have been introduced in order to address the suggestions of the Reviewer 2:

In this study the authors aim to evaluate in transplanted patients influence of socio-demographic factors and the time elapsed since liver transplantation on the functioning of patients.

The abstract and paper are difficult to read and need to be more concise and more flowing.

The abstract has been modified.

The research design is quite appropriate, and methods adequately described, but the monocentric design of study represent a limitation.

There are for me several points which need more clarification:

  • the authors should clarify the time frame for the study enrollment;

The missing information has been added.

  • the authors should clarify the exclusion criteria of the study and the number of transplanted patients who refused to be tested (they could be the patients with less compliant) ;

The missing information has been completed. Thirty-six patients were excluded because of their refusal to participate in the study.

  • The etiology of hepatic disease, especially alcoholic cirrhosis, before the transplantation might play a role in this setting and should be add in the analysis;
  • Are there patients with a psychiatric diagnosis? Or are these patients excluded from the study?

The missing information has been added―patients with a diagnosis of mental illness and those with alcoholic cirrhosis were excluded from the study.

Authors did not clarify how the research can have an impact on the clinical management of this patients.

  • The authors should add in the conclusions the limitations of the study: monocentric, no control group etc.

Supplemented in accordance with the Reviewer’s suggestions.

Kindest regards

Daria Schneider-Matyka

Round 2

Reviewer 1 Report

Dear authors,

With the operated changes and with the recognition that this study is hypothesis generating and cannot conclude on the matter, I can recommend the publication of your paper. Maybe the title is too strong (impact of...) for this type of study and you may consider adapting accordingly